# Are Emergent Abilities of Large Language Models a Mirage?

**Rylan Schaeffer**
Computer Science
Stanford University
rschaef@cs.stanford.edu

**Brando Miranda**
Computer Science
Stanford University
brando9@cs.stanford.edu

**Sanmi Koyejo**
Computer Science
Stanford University
sanmi@cs.stanford.edu

## Abstract

Recent work claims that large language models display *emergent abilities*: abilities not present in smaller-scale models that are present in larger-scale models. What makes emergent abilities intriguing is two-fold: their *sharpness*, transitioning seemingly instantaneously from not present to present, and their *unpredictability*, appearing at seemingly unforeseeable model scales. Here, we present an alternative explanation for emergent abilities: for a particular task and model family, when analyzing fixed model outputs, emergent abilities appear due to the researcher's choice of metric rather than due to fundamental changes in models with scale. Specifically, nonlinear or discontinuous metrics produce seemingly emergent abilities, whereas linear or continuous metrics produce smooth, continuous, predictable changes in model performance. We present our alternative explanation in a simple mathematical model, then test it in three complementary ways: we (1) make, test and confirm three predictions on the effect of metric choice using the InstructGPT/GPT-3 family on tasks with claimed emergent abilities; (2) make, test and confirm two predictions about metric choices in a meta-analysis of emergent abilities on the Beyond the Imitation Game Benchmark (BIG-Bench); and (3) show how to choose metrics to produce never-before-seen seemingly emergent abilities in multiple vision tasks across diverse deep network architectures. Via all three analyses, we provide evidence that emergent abilities disappear with different metrics or with better statistics, and may not be a fundamental property of scaling AI models.

## 1 Introduction

Emergent properties of complex systems have long been studied across disciplines, from physics to biology to mathematics. The idea of emergence was popularized by Nobel Prize-winning physicist P.W. Anderson's "More Is Different" [1], which argues that as the complexity of a system increases, new properties may materialize that cannot be predicted even from a precise quantitative understanding of the system's microscopic details. Recently, the idea of emergence gained significant attention in machine learning due to observations that large language models (LLMs) such as GPT [4], PaLM [7] and LaMDA [35] exhibit so-called "emergent abilities" [38, 9, 33, 4] (Fig. 1).

The term "emergent abilities of LLMs" was recently and crisply defined as "abilities that are not present in smaller-scale models but are present in large-scale models; thus they cannot be predicted by simply extrapolating the performance improvements on smaller-scale models" [38]. Such emergent abilities were first discovered in the GPT-3 family [4]. Subsequent work emphasized the discovery, writing that "[although model] performance is predictable at a general level, performance on a specific task can sometimes emerge quite unpredictably and abruptly at scale" [9]. These quotations collectively identify the two defining properties of emergent abilities in LLMs:

1. ***Sharpness***, transitioning seemingly instantaneously from not present to present

37th Conference on Neural Information Processing Systems (NeurIPS 2023).

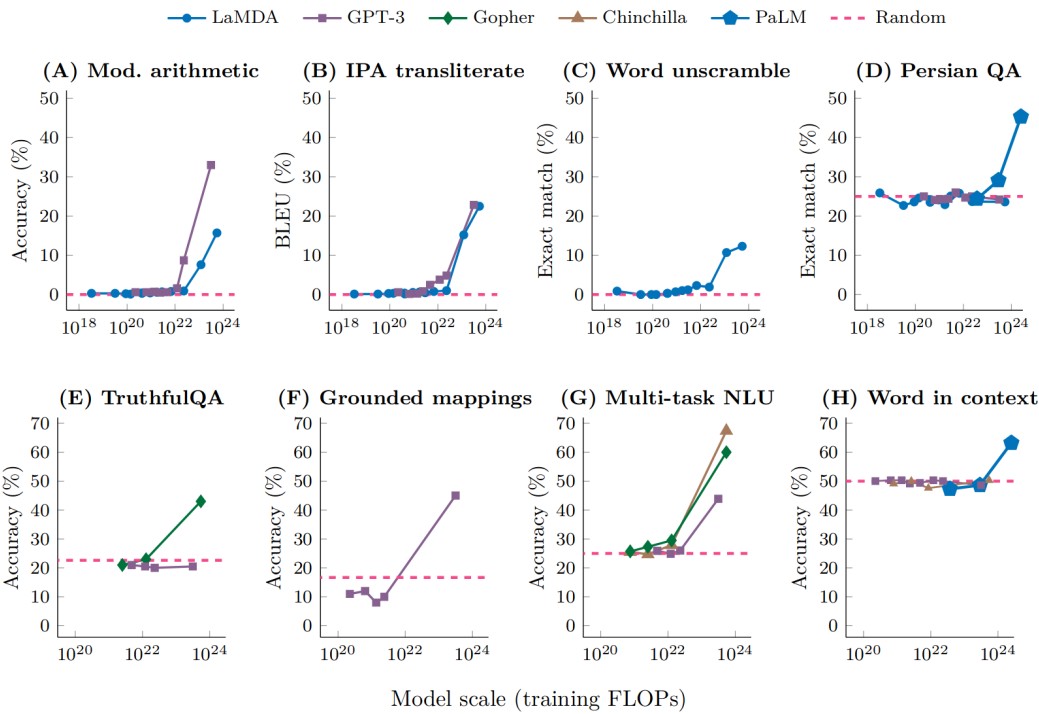

Figure 1: **Emergent abilities of large language models**. Model families display *sharp* and *unpredictable* increases in performance at specific tasks as scale increases. Source: Fig. 2 from [38].

    2. ***Unpredictability***, transitioning at seemingly unforeseeable model scales

These emergent abilities have garnered significant interest, raising questions such as: What controls *which* abilities will emerge? What controls *when* abilities will emerge? How can we make desirable abilities emerge faster, and ensure undesirable abilities never emerge? These questions are especially pertinent to AI safety and alignment, as emergent abilities forewarn that larger models might one day, without warning, acquire undesired mastery over dangerous capabilities [34, 12, 19, 20].

In this paper, we call into question the claim that LLMs possess emergent abilities, by which we specifically mean *sharp* and *unpredictable* changes in model outputs as a function of model scale on specific tasks. Our doubt stems from the observation that emergent abilities seem to appear only under metrics that nonlinearly or discontinuously scale any model's per-token error rate. For instance, as we later show, $> 92\%$ of emergent abilities on BIG-Bench tasks [33] (hand-annotated by [37]) appear under either of these two metrics:

$$\text{Multiple Choice Grade} \overset{\text{def}}{=} \begin{cases} 1 & \text{if highest probability mass on correct option} \\ 0 & \text{otherwise} \end{cases}$$

$$\text{Exact String Match} \overset{\text{def}}{=} \begin{cases} 1 & \text{if output string exactly matches target string} \\ 0 & \text{otherwise} \end{cases}$$

This raises the possibility of an alternative explanation for the origin of LLMs' emergent abilities: sharp and unpredictable changes might be induced by the researcher's choice of measurement, even though the model family's per-token error rate changes smoothly, continuously and predictably with increasing scale. Specifically, our alternative posits that emergent abilities are a mirage caused primarily by the researcher choosing a metric that nonlinearly or discontinuously deforms per-token error rates, and secondarily by possessing too few test data to accurately estimate the performance of smaller models, thereby causing smaller models to appear wholly unable to perform the task.

To communicate our alternative explanation, we present it as a simple mathematical model and demonstrate how it quantitatively reproduces the evidence offered in support of emergent abilities of LLMs. We then test our alternative explanation in three complementary ways:

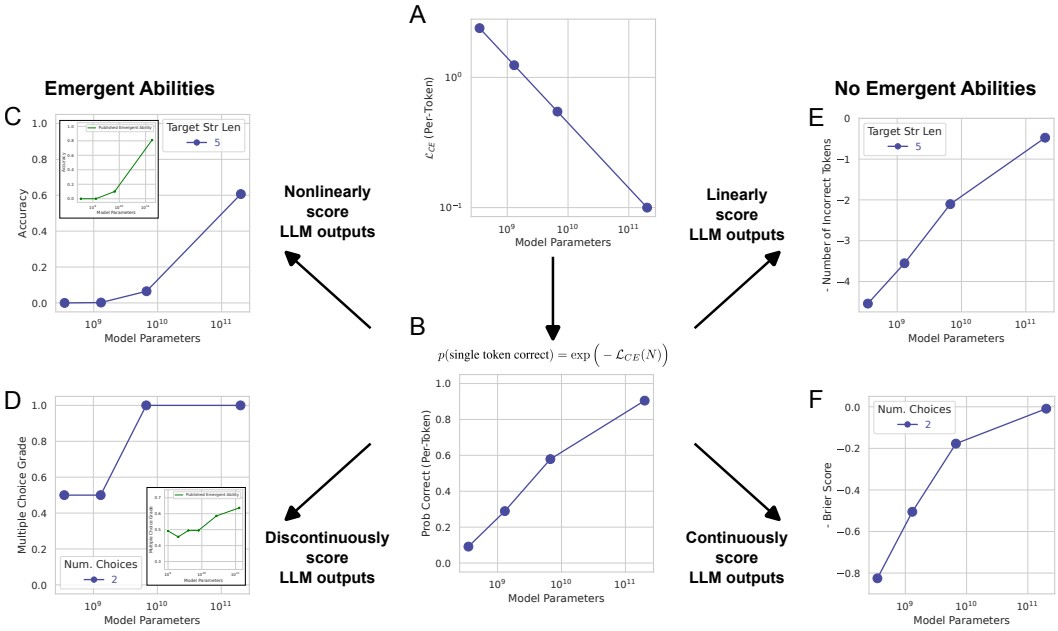

Figure 2: **Emergent abilities of large language models are created by the researcher's chosen metrics, not unpredictable changes in model behavior with scale.** (A) Suppose the per-token cross-entropy loss decreases monotonically with model scale, e.g., $\mathcal{L}_{CE}$ scales as a power law. (B) The per-token probability of selecting the correct token asymptotes towards 1. (C) If the researcher scores models' outputs using a nonlinear metric such as Accuracy (which requires a sequence of tokens to *all* be correct), the metric choice nonlinearly scales performance, causing performance to change sharply and unpredictably in a manner that qualitatively matches published emergent abilities (inset). (D) If the researcher instead scores models' outputs using a discontinuous metric such as Multiple Choice Grade (akin to a step function), the metric choice discontinuously scales performance, again causing performance to change sharply and unpredictably. (E) Changing from a nonlinear metric to a linear metric such as Token Edit Distance, scaling shows smooth, continuous and predictable improvements, ablating the emergent ability. (F) Changing from a discontinuous metric to a continuous metric such as Brier Score again reveals smooth, continuous and predictable improvements in task performance. Consequently, the observation of "emergent abilities" can be explained by the researcher's choice of metrics, and does not require fundamental changes in model family behavior on specific tasks with scale.

1. We make, test and confirm three predictions based on our alternative hypotheses using the InstructGPT [27] / GPT-3 [4] model family.

2. We meta-analyze published benchmarks [33, 38] to reveal that emergent abilities only appear for specific metrics, not for model families on particular tasks, and that changing the metric causes the emergence phenomenon to disappear.

3. We induce never-before-seen, seemingly emergent abilities in multiple architectures across various vision tasks by intentionally changing the metrics used for evaluation.

## 2   Alternative Explanation for Emergent Abilities

How might smooth, continuous, predictable changes in model family performance appear sharp and unpredictable? The answer is that the researcher's choice of a nonlinear or discontinuous metric can distort the model family's performance to appear sharp and unpredictable.

To expound, suppose that within a model family, the test loss falls smoothly, continuously, and predictably with the number of model parameters. One reason to believe this is the phenomenon known as neural scaling laws: empirical observations that deep networks exhibit power law scaling in the test loss as a function of training dataset size, number of parameters or compute [15, 32, 13, 18,

10, 14, 17, 39, 16, 8, 29]. For concreteness, suppose we have a model family of different numbers of parameters $N > 0$ and assume that each model's per-token cross entropy falls as a power law with the number of parameters $N$ for constants $c > 0, \alpha < 0$ (Fig. 2A):

$$\mathcal{L}_{CE}(N) = \left(\frac{N}{c}\right)^{\alpha}$$

To be clear, we do not require this particular functional form to hold; rather, we use it for illustrative purposes. Let $V$ denote the set of possible tokens, $p$ denote the true but unknown probability distribution, and $\hat{p}_N$ denote the $N$-parameter model's predicted probability distribution. The per-token cross entropy as a function of number of parameters $N$ is:

$$\mathcal{L}_{CE}(N) \overset{\text{def}}{=} -\sum_{v \in V} p(v) \log \hat{p}_N(v)$$

In practice, $p$ is unknown, so we substitute a one-hot distribution of the observed token $v^*$:

$$\mathcal{L}_{CE}(N) = -\log \hat{p}_N(v^*)$$

A model with $N$ parameters then has a per-token probability of selecting the correct token (Fig. 2B):

$$p(\text{single token correct}) = \exp\Big(-\mathcal{L}_{CE}(N)\Big) = \exp\Big(-(N/c)^{\alpha}\Big)$$

Suppose the researcher then chooses a metric that requires selecting $L$ tokens correctly. For example, our task might be $L$-digit integer addition, and a model's output is scored 1 if all $L$ output digits exactly match all target digits with no additions, deletions or substitutions, 0 otherwise. If the probability each token is correct is independent[1], the probability of scoring 1 is:

$$\text{Accuracy}(N) \approx p_N(\text{single token correct})^{\text{num. of tokens}} = \exp\Big(-(N/c)^{\alpha}\Big)^{L}$$

This choice of metric nonlinearly scales performance with increasing token sequence length. When plotting performance on a linear-log plot, one sees a sharp, unpredictable emergent ability on longer sequences (Fig. 2C) that closely matches claimed emergent abilities (inset). What happens if the researcher switches from a nonlinear metric like Accuracy, under which the per-token error rate scales geometrically in target length (App. A.3), to an approximately linear metric like Token Edit Distance, under which the per-token error rate scales quasi-linearly in target length (App. A.2)?

$$\text{Token Edit Distance}(N) \approx L\Big(1 - p_N(\text{single token correct})\Big) = L\Big(1 - \exp\big(-(N/c)^{\alpha}\big)\Big)$$

The linear metric reveals smooth, continuous, predictable changes in model performance (Fig. 2E). Similarly, if the researcher uses a discontinuous metric like Multiple Choice Grade, the researcher can find emergent abilities (Fig. 2D), but switching to a continuous metric like Brier Score removes such abilities (Fig. 2F). In summary, sharp and unpredictable changes with increasing scale can be fully explained by three interpretable factors: (1) the researcher choosing a metric that nonlinearly or discontinuously scales the per-token error rate, (2) having insufficient resolution to estimate model performance in the smaller parameter regime, with resolution[2] set by 1/test dataset size, and (3) insufficiently sampling the larger parameter regime.

## 3 Analyzing InstructGPT/GPT-3's Emergent Arithmetic Abilities

Previous papers prominently claimed the GPT [4, 27] family[3] displays emergent abilities at integer arithmetic tasks [9, 33, 38] (Fig. 1A). We chose these tasks as they were prominently presented

---

[1] While the independence assumption is not true, the approximation yields results qualitatively matching the observed emergence claims.

[2] Resolution is defined as "The smallest interval measurable by a scientific instrument; the resolving power."

[3] As of 2023-03-15, 4 models with 350M, 1.3B, 6.7B, 175B parameters are available via the OpenAI API.

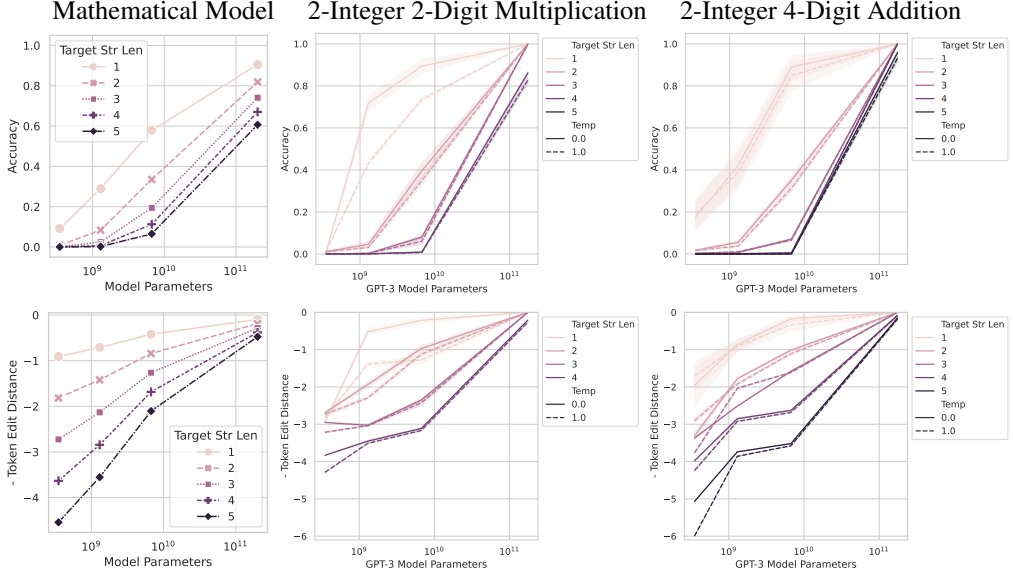

Figure 3: **Claimed emergent abilities evaporate upon changing the metric.** Top: When performance is measured by a nonlinear metric (e.g., Accuracy), the InstructGPT/GPT-3 [4, 27] family's performance appears sharp and unpredictable on longer target lengths. Bottom: When performance is instead measured by a linear metric (e.g., Token Edit Distance), the family exhibits smooth, predictable performance improvements.

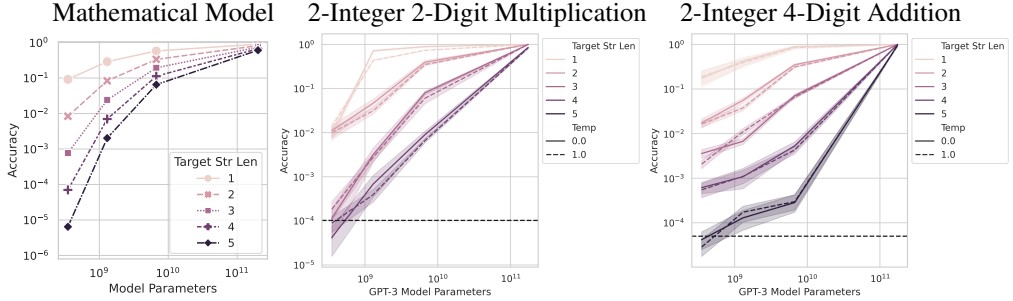

Figure 4: **Claimed emergent abilities evaporate upon using better statistics.** Based on the predictable effect Accuracy has on performance, measuring performance requires high resolution. Generating additional test data increases the resolution and reveals that even on Accuracy, the InstructGPT/GPT-3 family's [4, 27] performance is above chance and improves in a smooth, continuous, predictable manner that qualitatively matches the mathematical model.

[4, 9, 33, 38], and we focused on the GPT family due to it being publicly queryable. As explained mathematically and visually in Sec. 2, our alternative explanation makes three predictions:

1. Changing the metric from a nonlinear or discontinuous metric (Fig. 2CD) to a linear or continuous metric (Fig. 2 EF) should reveal smooth, continuous, predictable performance improvement with model scale.

2. For nonlinear metrics, increasing the resolution of measured model performance by increasing the test dataset size should reveal smooth, continuous, predictable model improvements *commensurate with the predictable nonlinear effect of the chosen metric*.

3. Regardless of metric, increasing the target string length should predictably affect the model's performance as a function of the length-1 target performance: approximately geometrically for accuracy and approximately quasilinearly for token edit distance.

To test these predictions, we collected outputs from the InstructGPT/GPT-3 family on two tasks: 2-shot multiplication between two 2-digit integers and 2-shot addition between two 4-digit integers.

**Prediction: Emergent Abilities Disappear With Different Metrics** On both arithmetic tasks, the GPT family displays emergent abilities if the target has 4 or 5 digits and if the metric is Accuracy (Fig. 3, top) [4, 9, 38]. However, if one changes from nonlinear Accuracy to linear Token Edit Distance *while keeping the models' outputs fixed*, the family's performance smoothly, continuously and predictably improves with increasing scale (Fig. 3, bottom). This confirms our first prediction and supports our alternative explanation that the observation of emergent abilities can be explained by the researcher's choice of metric, *not changes in the model family's outputs*. We also observe that under Token Edit Distance, increasing the length of the target string from 1 to 5 predictably decreases the family's performance in an approximately quasilinear manner, confirming the first half of our third prediction.

**Prediction: Emergent Abilities Disappear With Better Statistics** We next tested our second prediction: that even on nonlinear metrics such as accuracy, smaller models do not have zero accuracy, but rather have non-zero above-chance accuracy *commensurate with choosing to use accuracy as the metric*. In order to accurately measure models' accuracy, we increased the resolution by generating additional test data, and found that on both arithmetic tasks, all models in the InstructGPT/GPT-3 family achieve above-chance accuracy (Fig. 4). This confirms our second prediction. We also observe that as the target string length increases, the accuracy falls approximately geometrically with the length of the target string, confirming the second half of our third prediction. These results additionally demonstrate that the researcher's choice of metric has the effect that one should predict accuracy to have, i.e., geometric decay with the target length.

# 4 Meta-Analysis of Claimed Emergent Abilities

Analyzing the GPT family is possible because the models are publicly queryable. However, at the time of this analysis, other model families claimed to exhibit emergent abilities are not publicly queryable, nor are their generated outputs publicly available, meaning we are limited to analyzing the published results themselves [9, 38, 37]. Our alternative explanation makes two predictions.

1. At the "population level" of Task-Metric-Model Family triplets, emergent abilities should appear predominantly on specific *metrics*, not *task-model family* pairs, and specifically with nonlinear and/or discontinuous metrics.

2. On individual Task-Metric-Model Family triplets that display an emergent ability, changing the metric to a linear and/or continuous metric should remove the emergent ability.

To test these predictions, we used claimed emergent abilities on BIG-Bench [33, 38] due to the benchmark being pertinent and publicly available.

**Prediction: Emergent Abilities Should Appear with Metrics, not Task-Model Families** If emergent abilities are real, one should expect task-model family pairs to show emergence for all reasonable metrics. However, if our alternative explanation is correct, we should expect emergent abilities to appear only under certain metrics. To test this, we analyzed on which metrics emergent abilities appear. To determine whether a task-metric-model family triplet exhibits a possible emergent ability, we used a metric from previous work [33]. Letting $y_i \in \mathbb{R}$ denote model performance at model scales $x_i \in \mathbb{R}$, sorted such that $x_i < x_{i+1}$, the emergence score is:

$$\text{Emergence Score}\Big(\big\{(x_n, y_n)\big\}_{n=1}^{N}\Big) \stackrel{\text{def}}{=} \frac{\text{sign}(\arg\max_i y_i - \arg\min_i y_i)(\max_i y_i - \min_i y_i)}{\sqrt{\text{Median}(\{(y_i - y_{i-1})^2\}_i)}}$$

We found that most metrics used in BIG-Bench have *zero* task-model family pairs that exhibit emergent abilities: of the 39 preferred metrics in BIG-Bench, at most 5 display emergence (Fig. 5A). Many of the 5 are nonlinear and/or discontinuous, e.g., Exact String Match, Multiple Choice Grade, ROUGE-L-Sum (App. A.4). Notably, because BIG-Bench often scores models on tasks using multiple metrics, the *lack* of emergent abilities under other metrics suggests that emergent abilities do not appear when model outputs are scored using other metrics.

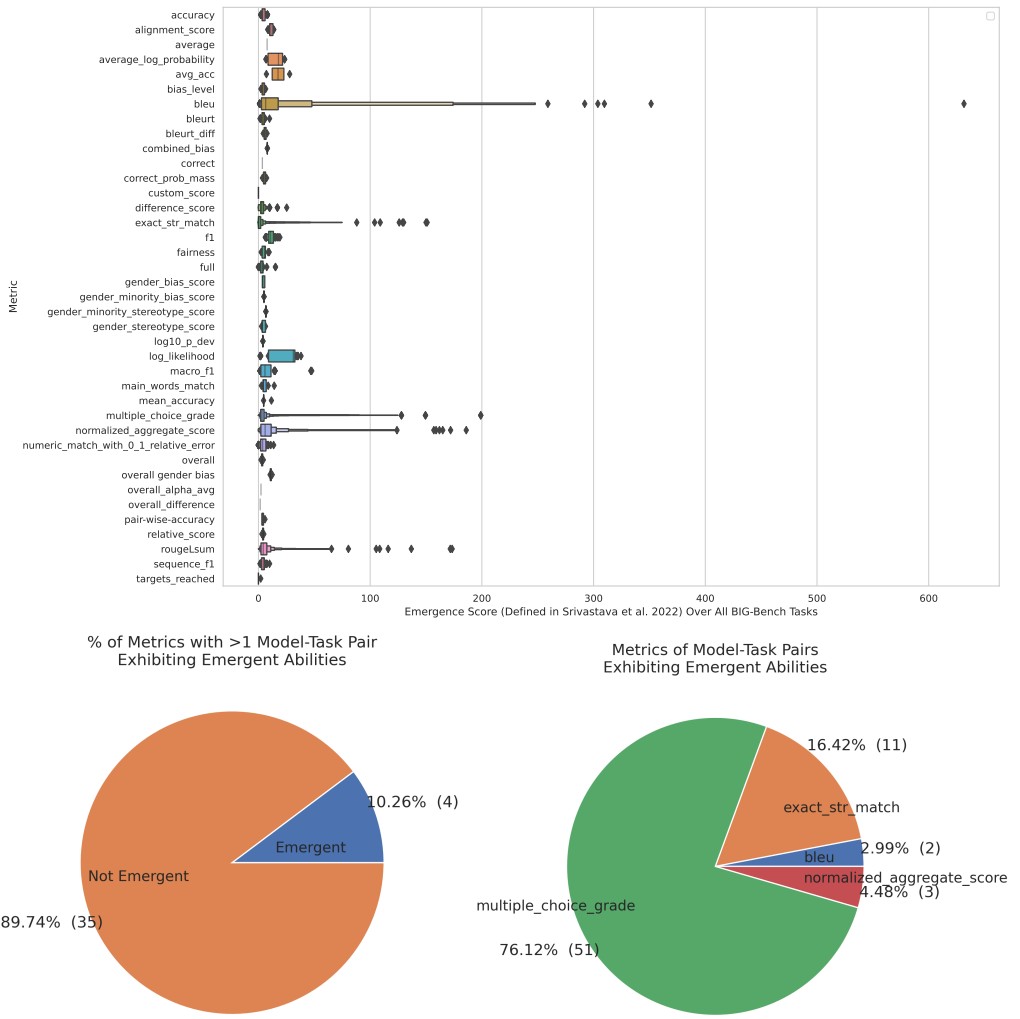

Figure 5: **Emergent abilities appear only for specific metrics, not task-model families.** (A) *Possible* emergent abilities appear with *at most* 5 out of 39 BIG-Bench metrics. (B) Hand-annotated data by [37] reveal emergent abilities appear only under 4 preferred metrics. (C) $> 92\%$ of emergent abilities appear under one of two metrics: Multiple Choice Grade and Exact String Match.

Because emergence score only *suggests* emergence, we also analyzed hand-annotated task-metric-model family triplets [37], which revealed emergent abilities appear with $4/39$ metrics (Fig. 5B), and 2 metrics account for $> 92\%$ of claimed emergent abilities (Fig. 5C): Multiple Choice Grade and Exact String Match. Multiple Choice Grade is discontinuous, and Exact String Match is nonlinear.

**Prediction: Changing Metric Removes Emergent Abilities**   To test our second prediction, we focused on the LaMDA family [35] because its outputs are available through BIG-Bench. We identified tasks on which LaMDA displays emergent abilities with Multiple Choice Grade, then asked whether LaMDA still displays emergent abilities on the same tasks with a different BIG-Bench metric: Brier Score [3]. Brier Score is a strictly proper scoring rule for predictions of mutually exclusive outcomes; for a binary outcome, the Brier Score simplifies to the squared error between 1 and the model's probability mass on the outcome. LaMDA's emergent abilities on the discontinuous Multiple Choice Grade disappeared when we changed the metric to the continuous Brier Score (Fig. 6). These results support our alternative explanation that emergent abilities are induced by the chosen metric.

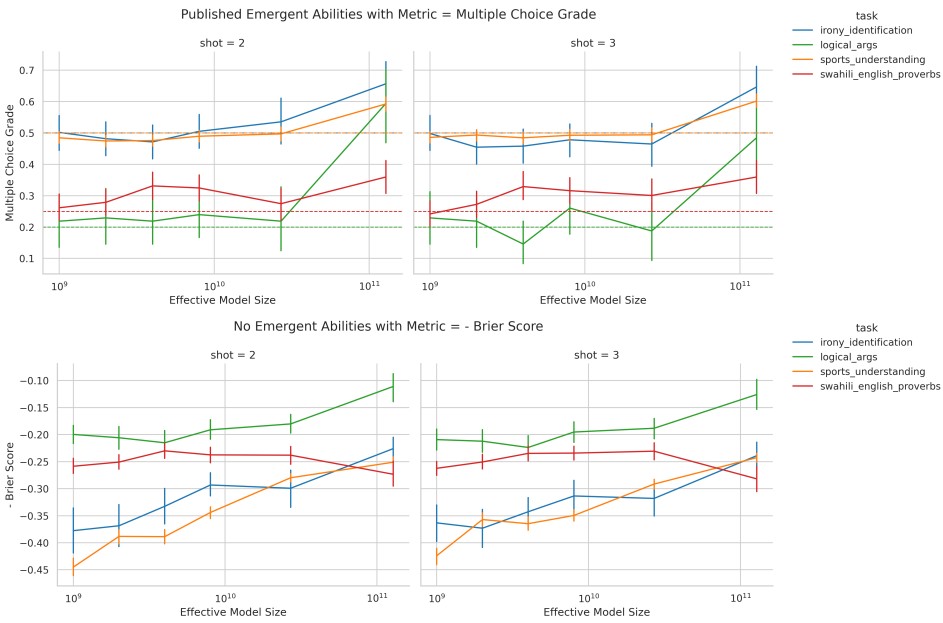

Figure 6: **Changing the metric when evaluating task-model family pairs causes emergent abilities to disappear.** Top: The LaMDA model family displays emergent abilities when measured under the discontinuous Multiple Choice Grade. Bottom: The LaMDA model family's emergent abilities disappear when measured under a continuous BIG-Bench metric: Brier Score.

## 5   Inducing Emergent Abilities in Networks on Vision Tasks

To demonstrate how emergent abilities can be induced by the researcher's choice of metric, we show how to produce emergent abilities in deep networks of various architectures: fully connected, convolutional, self-attentional. We focus on vision tasks because abrupt transitions in vision models' capabilities have not been observed to the best of our knowledge; this is one reason why emergence in large language models is considered so interesting. For the convolutional example, see App. B.

**Emergent Reconstruction of CIFAR100 Natural Images by Nonlinear Autoencoders**   We first induce an emergent ability to reconstruct images in shallow (i.e., single hidden layer) nonlinear autoencoders trained on CIFAR100 natural images [21]. To emphasize that the sharpness of the metric is responsible for emergent abilities, and to show that sharpness extends to metrics beyond Accuracy, we intentionally define a discontinuous metric that measures a network's ability to reconstruct a dataset as the average number of test data with squared reconstruction error below cutoff $c$:

$$\text{Reconstruction}_c\left(\{x_n\}_{n=1}^{N}\right) \stackrel{\text{def}}{=} \frac{1}{N}\sum_n \mathbb{I}\Big[||x_n - \hat{x}_n||^2 < c\Big], \tag{1}$$

where $\mathbb{I}(\cdot)$ denotes an indicator variable and $\hat{x}_n$ is the autoencoder's reconstruction of $x_n$. The autoencoder family displays smoothly decreasing squared reconstruction error as the number of bottleneck units increases (Fig. 7B). Under our newly defined Reconstruction$_c$ metric and for particular choices of $c$, the autoencoder family exhibits a sharp and seemingly unpredictable image reconstruction ability (Fig. 7C) that qualitatively matches published emergent abilities (Fig. 7A).

**Emergent Classification of Omniglot Characters by Autoregressive Transformers**   We next induce emergent abilities in Transformers [36] trained to autoregressively classify Omniglot hand-written characters [22], in a setup inspired by recent work [6]: Omniglot images are embedded by convolutional layers, then sequences of embedded image-image class label pairs are fed into decoder-only transformers. We measure image classification performance on sequences of length $L \in [1, 5]$, again via *subset accuracy*: 1 if all $L$ images are classified correctly (Fig. 8B), 0 otherwise. Causal transformers display a seemingly emergent ability to correctly classify Omniglot handwritten characters (Fig. 8C) that qualitatively matches published emergent abilities (Fig. 8A).

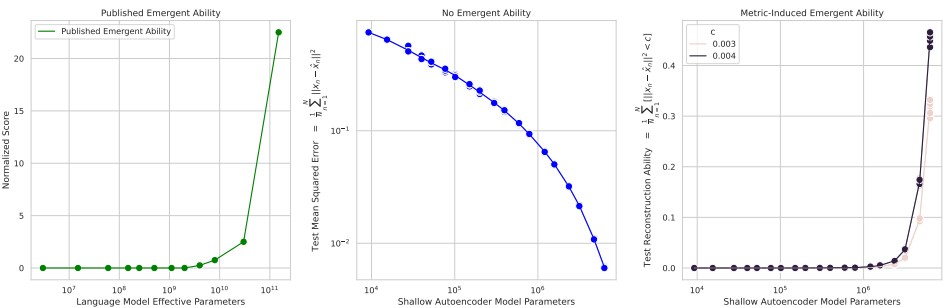

Figure 7: **Induced emergent reconstruction ability in shallow nonlinear autoencoders.** (A) A published emergent ability at the BIG-Bench Periodic Elements task [33]. (B) Shallow nonlinear autoencoders trained on CIFAR100 [21] display smoothly decreasing mean squared reconstruction error. (C) Using a newly defined Reconstruction$_c$ metric (Eqn. 1) induces an unpredictable change.

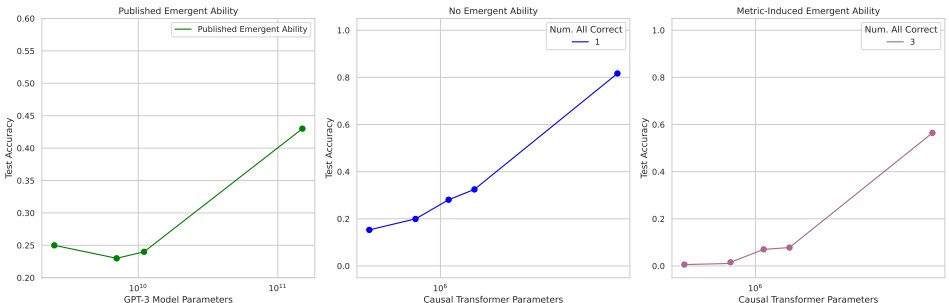

Figure 8: **Induced emergent classification ability in autoregressive Transformers.** (A) A published emergent ability on the MMLU benchmark [9]. (B) Autoregressive transformers trained to classify Omniglot images display increasing accuracy with increasing scale. (C) When accuracy is redefined as classifying *all* images correctly, a seemingly emergent ability appears.

## 6 Limitations

This paper has several limitations. First, nothing in this paper should be interpreted as claiming that large language models *cannot* display emergent abilities; rather, our message is that some previously claimed emergent abilities appear to be mirages induced by researcher analyses. Second, our experiments and analyses are limited because some LLMs with claimed emergent abilities (e.g., PaLM 1, Gopher, Chinchilla) are private and not queryable at the time of our analysis. Lastly, the best metric(s) arguably depends on human preferences, which may exhibit qualitatively different behavior; we are unaware of studies quantifying whether human judgment is thresholded in an "emergent" way.

## 7 Related Work

Srivastava et al. [33] observed that while accuracy at a particular task can empirically appear sharp and unpredictable, cross-entropy does not appear so; the authors then discussed whether emergent abilities may be partially attributed to the metric. Our paper converts their discussion into precise predictions, then quantitatively tests the predictions to reveal metric choice is possibly responsible for some claimed emergent abilities; well-known and widely-used metrics (including metrics used by [33]) capture graded improvements; emergent abilities do not appear only on tasks involving multiple steps, such as the discontinuous Multiple Choice Grade; metric choice can be used to induce emergent abilities in a novel domain (vision) in diverse architectures and tasks.

Alternative explanations exist for the origin of emergent abilities. Caballero et al. [5] explain emergence by assuming a piece-wise power law functional form; under this view, emergent abilities are real, caused by a "break" (or possibly multiple breaks) in the governing power law. In contrast, our work suggests that emergent abilities can be induced by the researcher under a single power law. Both

explanations could be true: some emergent abilities might genuinely be abruptly appearing, whereas some emergent abilities might be attributable to the metric. Michaud et al. [28] posits that language modeling data might be comprised of discrete subtasks ("quanta") that networks learn; if larger networks have greater capacity, and are thus more capable of learning more of these quanta, then if some downstream task requires a network to learn some combination of quanta, larger networks are more likely to have all the requisite capabilities and thus are capable of performing this downstream task. We think that this is a very interesting hypothesis. Whether language modeling data can or should be understood from this quantization perspective, and whether these quanta indeed are the origin of emergent abilities, are really exciting questions that we think merit more study.

## 8 Discussion

Our paper presents an alternative explanation for the claimed emergent abilities of large language models. For a fixed task and a fixed model family, the researcher can choose a metric to create an emergent ability or choose a metric to ablate an emergent ability. Ergo, *emergent abilities may be creations of the researcher's choices, not a fundamental property of the model family on the specific task.*

Our work has several implications. Firstly, a task and a metric are distinct and meaningful choices when constructing a benchmark. Secondly, when choosing metric(s), if the goal is to accurately predict scaling behavior, then one should consider the interplay between cross-entropy, transformations, and resolution-limited evaluations so that one isn't surprised. As a corollary, continuous/linear metrics are probably better for accurate scaling forecasts, but if discontinuous/nonlinear metrics are preferred, then one may need a lot of data for sufficient resolution to accurately measure performance. The key is thinking through the consequences of one's choices! Thirdly, when making claims about capabilities of large models, including proper controls is critical. In this particular setting, emergent abilities claims are possibly infected by a failure to control for multiple comparisons. In BIG-Bench alone, there are $\geq 220$ tasks, $\sim 40$ metrics per task, $\sim 10$ model families, for a total of $\sim 10^6$ task-metric-model family triplets, meaning the probability that *no* task-metric-model family triplet exhibits an emergent ability by random chance might be small. Fourthly, scientific progress can be hampered when models and their outputs are not made available for independent scientific investigation.

## 9 Contributions

RS conceived of the research direction collected data, ran experiments, and analyzed results. SK supervised and guided the project. BM also provided guidance. All authors helped write the manuscript.

## 10 Acknowledgements

This work is partially supported by the National Science Foundation under grants No. 2046795, 1934986, 2205329, NIH 1R01MH116226-01A, NIFA award 2020-67021-32799, the Alfred P. Sloan Foundation, and Google Inc. RS is partially supported by a Stanford Data Science Scholarship and BM is partially supported by a Stanford School of Engineering Fellowship and a Stanford EDGE Scholar Fellowship. We thank our colleagues Professor Tatsunori Hashimoto, Eric Han, Max Lamparth, Mikail Khona, Kateryna Pistunova, Victor Lecomte, and Zane Durante for discussing our findings with us and providing much-appreciated feedback.

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

# A  Approximate Behavior of Metrics on Sequential Data

How do different metrics behave when used to measure autoregressive model outputs? Precisely answering this question is tricky and possibly analytically unsolvable, so we provide an approximate answer here.

Notationally, we consider $N$ test data of length $L$ (here, length is measured in tokens) with targets denoted $t_n \stackrel{\text{def}}{=} (t_{n1}, t_{n2}, ... t_{nL})$, the autoregressive model has a true-but-unknown per-token error probability of $\epsilon \in [0, 1]$ and the model outputs prediction $\hat{t}_n \stackrel{\text{def}}{=} (\hat{t}_{n1}, \hat{t}_{n2}, ... \hat{t}_{nL})$. This assumes that the model's per-token error probability is constant, which is empirically false, but modeling the complex dependencies of errors is beyond our scope.

## A.1  Per-Token Error Probability is Resolution-Limited

Note that because we have $N$ test data, each of length $L$, our resolution for viewing the per-token error probability $\epsilon$ is limited by $1/NL$. Here, resolution refers to "the smallest interval measurable by a scientific instrument; the resolving power." To explain what resolution means via an example, suppose one wants to measure a coin's probability of yielding heads. After a single coin flip, only two outcomes are possible (H, T), so the resolution-limited probability of heads is either $0$ or $1$. After two coin flips, four outcomes are possible (HH, HT, TH, TT), so the resolution-limited probability of heads is now one of $0, 0.5, 1$. After $F$ coin flips, we can only resolve the coin's probability of yielding heads up to $1/F$. Consequently, we introduce a resolution-limited notation:

$$a_b \stackrel{\text{def}}{=} a \text{ rounded to the nearest integer multiple of } 1/b \tag{2}$$

## A.2  Token Edit Distance

We first consider an adaptation of the Levenshtein (string edit) distance for models that function on tokens rather than characters, an adaptation we term the *token edit distance*. The token edit distance between two token sequences $t_n, \hat{t}_n$ is defined as the integer number of additions, deletions or substitutions necessary to transform $t_n$ into $\hat{t}_n$ (or vice versa).

$$\text{Token Edit Distance}(t_n, \hat{t}_n) \stackrel{\text{def}}{=} \text{Num Substitutions} + \text{Num. Additions} + \text{Num. Deletions} \tag{3}$$

$$= \sum_{\ell=1}^{L} \mathbb{I}[t_{n\ell} \neq \hat{t}_{n\ell}] + \text{Num. Additions} + \text{Num. Deletions} \tag{4}$$

$$\geq \sum_{\ell=1}^{L} \mathbb{I}[t_{n\ell} \neq \hat{t}_{n\ell}] \tag{5}$$

The expected token edit distance is therefore:

$$\mathbb{E}[\text{Token Edit Distance}(t_n, \hat{t}_n)] \geq \mathbb{E}[\sum_{\ell=1}^{L} \mathbb{I}[t_{n\ell} \neq \hat{t}_{n\ell}]] \tag{6}$$

$$= \sum_{\ell=1}^{L} p(t_{n\ell} \neq \hat{t}_{n\ell}) \tag{7}$$

$$\approx L(1 - \epsilon) \tag{8}$$

The resolution-limited expected token edit distance is therefore:

$$\mathbb{E}[\text{Token Edit Distance}(t_n, \hat{t}_n)]_{NL} \geq L\left(1 - \epsilon_{NL}\right) \tag{9}$$

From this, we see that the expected token edit distance scales approximately linearly with the resolution-limited per-token probability. The real rate is slightly higher than linear because additions

and deletions contribute an additional non-negative cost, but modeling this requires a model of how likely the model is to overproduce or underproduce tokens, which is something we do not currently possess.

## A.3 Accuracy

$$\text{Accuracy}(t_n, \hat{t}_n) \overset{\text{def}}{=} \mathbb{I}[\text{No additions}] \, \mathbb{I}[\text{No deletions}] \prod_{l=1}^{L} \mathbb{I}[t_{nl} = \hat{t}_{nl}] \tag{10}$$

$$\approx \prod_{l=1}^{L} \mathbb{I}[t_{nl} = \hat{t}_{nl}] \tag{11}$$

As with the Token Edit Distance (App. A.2), we ignore how likely the language model is to overproduce or underproduce tokens because we do not have a good model of this process. Continuing along,

$$\mathbb{E}[\log \text{Accuracy}] = \sum_{l} \mathbb{E}[\log \mathbb{I}[t_{nl} = \hat{t}_{nl}]] \tag{12}$$

$$\leq \sum_{l} \log \mathbb{E}[\mathbb{I}[t_{nl} = \hat{t}_{nl}]] \tag{13}$$

$$\approx L \log(1 - \epsilon) \tag{14}$$

Taking an approximation that would make most mathematicians cry:

$$\mathbb{E}[\text{Accuracy}] \approx \exp(\mathbb{E}[\log \text{Accuracy}]) \tag{15}$$

$$= (1 - \epsilon)^L \tag{16}$$

$$\tag{17}$$

This reveals that accuracy **approximately** falls geometrically with target token length. The resolution-limited expected accuracy is therefore:

$$\mathbb{E}[\text{Accuracy}]_{NL} = (1 - \epsilon)^L{}_{NL} \tag{18}$$

From this we can see that choosing a nonlinear metric like Accuracy is affected significantly more than a linear metric by limited resolution because Accuracy forces one to distinguish quantities that decay rapidly.

## A.4 ROUGE-L-Sum

Another BIG-Bench metric [33] is ROUGE-L-Sum [25], a metric based on the longest common subsequence (LCS) between two sequences. Section 3.2 of [25] gives the exact definition, but the key property is that ROUGE-L-Sum measures the "union" LCS, which means "stitching" together LCSs across the candidate and multiple references. As explained in the original paper [25]: if the candidate sequence is $c = w_1 w_2 w_3 w_4 w_5$, and if there are two reference sequences $r_1 = w_1 w_2 w_6 w_7 w_8$ and $r_2 = w_1 w_3 w_8 w_9 w_5$, then $LCS(r_1, c) = w_1 w_2$ and $LCS(r_2, c) = w_1 w_3 w_5$, then the union LCS of $c, r_1, r_2$ is $w_1 w_2 w_3 w_5$, with length 4. Intuitively, this disproportionately benefits models with smaller error rates because their mistakes can be "stitched" across multiple references; this is confirmed in Monte Carlo simulation (Fig. 9).

## A.5 BLEU

Yet another BIG-Bench metric [33] is BLEU [30], a metric based on shared n-grams between the generated string and reference strings. BLEU is also a discontinuous *and* nonlinear metric for several

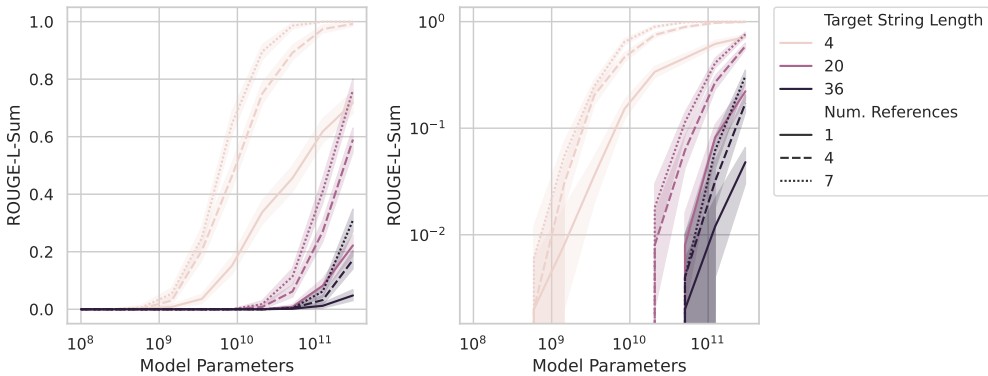

Figure 9: **ROUGE-L-Sum is a sharp metric.** Simulations show that as the per-token error probability slightly increases (e.g. from 0.05 to 0.1), the ROUGE-L-Sum metric falls sharply.

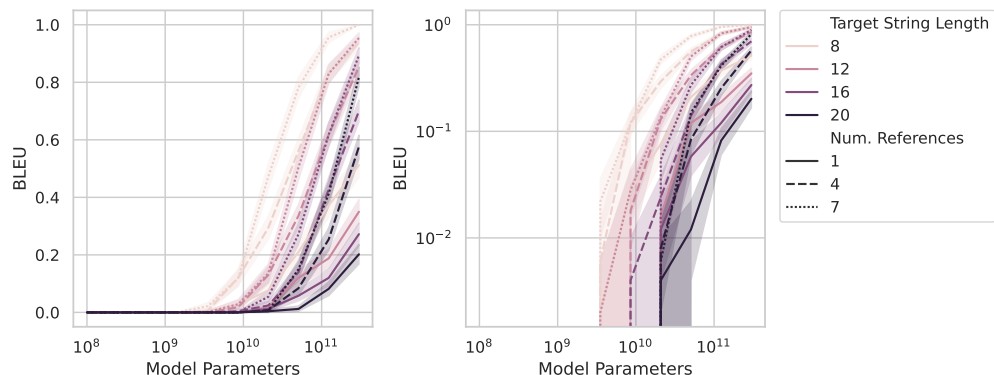

Figure 10: **BLEU is a sharp metric.** Simulations show that as the per-token error probability slightly increases (e.g. from 0.05 to 0.1), the ROUGE-L-Sum metric falls sharply.

reasons. For an explanation of its discontinuity, consider bleu.compute(predictions=["hello there general"], references=[["hello there general"]]). At first glance, this might seem like it should also result in a BLEU score of 1.0 since the prediction matches the reference. However, the issue here is the absence of longer n-grams. For the unigrams, bigrams, and trigrams, the precision is 1.0 since they match perfectly. However, for the 4-grams, there are none in both the candidate and the reference. This results in a precision of 0 for the 4-grams because the BLEU score takes the geometric mean of the n-gram precisions, meaning any 0 in the set will make the entire product 0. Hence, despite the match in unigrams, bigrams, and trigrams, the absence of 4-grams results in a BLEU score of 0.0. This behavior of BLEU has been a point of criticism, as short sentences or those with fewer n-grams than the maximum considered (often 4) can yield scores that are counter-intuitive. This is confirmed in Monte Carlo simulations (Fig. 10)

## B    Inducing Emergent Abilities in Networks on Vision Tasks

### B.1    Emergent Classification of MNIST Handwritten Digits by Convolutional Networks

We begin by inducing an emergent classification ability in a LeNet convolutional neural network family [24], trained on the MNIST handwritten digits dataset [23]. This family displays smoothly increasing test accuracy as the number of parameters increases (Fig. 11B). To emulate the accuracy metric used by emergence papers [9, 38, 33], we use *subset accuracy*: 1 if the network classifies $K$ out of $K$ (independent) test data correctly, 0 otherwise. Under this definition of accuracy, the model family displays an "emergent" ability to correctly classify sets of MNIST digits as $K$ increases from 1 to 5, especially when combined with sparse sampling of model sizes (Fig. 11C). This convolutional

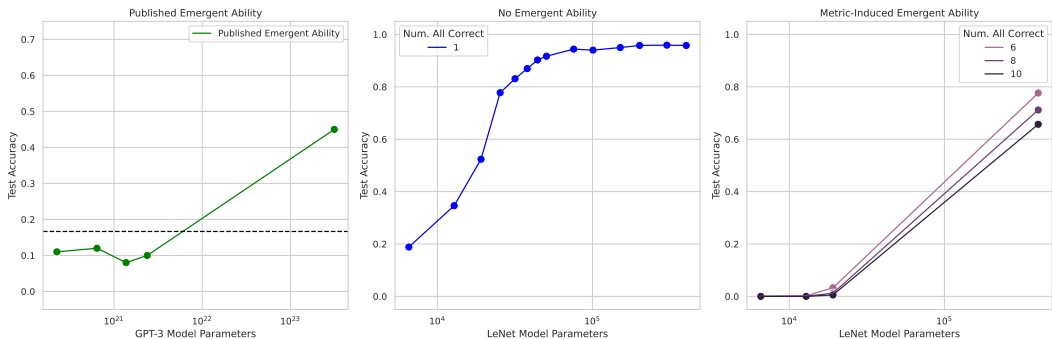

Figure 11: **Induced emergent MNIST classification ability in convolutional networks.** (A) A published emergent ability from the BIG-Bench Grounded Mappings task [38]. (B) LeNet trained on MNIST [23] displays a predictable, commonplace sigmoidal increase in test accuracy as model parameters increase. (C) When accuracy is redefined as correctly classifying $K$ out of $K$ independent test data, this newly defined metric induces a seemingly unpredictable change.

family's emergent classification ability qualitatively matches published emergent abilities, e.g., at the BIG-Bench Grounded Mappings task [38] (Fig. 11A).

## C    Relationship Between Emergent Abilities and Grokking

Emergent abilities [4, 9, 33, 38] are sometimes compared with grokking [31, 26, 2, 11], a phenomenon whereby a single model will, over the course of learning, achieve high training accuracy and only much later achieve high test accuracy. There are several differences between grokking and emergent abilities:

1. Grokking is primarily studied within a single model, whereas emergent abilities are studied within a model family (i.e., multiple models).

2. Grokking occurs with increasing gradient steps, whereas emergent abilities occur with increasing model scale, typically measured in parameters or effective parameters (although more recently compute).

3. Grokking explicitly studies a discrepancy between the model's train and test behavior, whereas emergent abilities (to the best of our knowledge) do not present separate train & test curves.

4. Grokking is primarily studied on toy "algorithmic" tasks in small networks, whereas emergent abilities are often studied on benchmark NLP tasks in large language models.

