# OpenReview forum: "Are Emergent Abilities of Large Language Models a Mirage?"
_NeurIPS.cc/2023/Conference — NeurIPS 2023 oral_

### Official Review · Reviewer_ccWP · 2023-07-05

**Soundness:** 4 excellent
**Presentation:** 4 excellent
**Contribution:** 3 good
**Rating:** 7
**Confidence:** 4

**Summary:**

The authors argue that purported emergent abilities of language models are mostly explained by the use of specific metrics, including non-linear or discontinuous scoring, and evaporate when linear or continuous scoring metrics are applied. The authors (1) present a toy model to explain emergence, (2) demonstrate that most abilities previously reported as emergent no longer appear emergent under an alternative metric, and then synthetically cause the appearance of emergence in an image reconstruction task.

**Strengths:**

- Simply and clearly addresses the enigmatic phenomenon of emergence

- Formalizes and quantitatively tests the hypothesis of [1] that emergence is due in part to metric

- Compelling toy model to illustrate how emergence can appear with non-linear metric (accuracy) but vanishes with linear metric (token edit distance)

- Convincingly demonstrates that emergence is (at least often) a result of the metric

**Weaknesses:**

- As the authors point out in their related work, the idea that emergence appears due to the metric is not novel, and was proposed by [1], so the novelty of this paper lies in the quantitative evaluation of this hypothesis

- There has been work on examining emergence and progress measures in similar contexts, e.g. grokking, which should be addressed in the related work [2–5]

- There was no discussion of the relationship to alternative hypotheses. The authors mention in the related work multiple papers that suggest that abilities do, in fact, emerge with scale, but the authors should explain how their hypothesis relates

- There was no discussion of the BLEU score metric, even though there are many tasks that have a very high score with this metric

- Minor: a lot of the figures are small and therefore hard to read

[1] Srivastava, Aarohi, et al. "Beyond the imitation game: Quantifying and extrapolating the capabilities of language models." arXiv preprint arXiv:2206.04615 (2022).

[2] Power, Alethea, et al. "Grokking: Generalization beyond overfitting on small algorithmic datasets." arXiv preprint arXiv:2201.02177 (2022).

[3] Liu, Ziming, Eric J. Michaud, and Max Tegmark. "Omnigrok: Grokking beyond algorithmic data." arXiv preprint arXiv:2210.01117 (2022).

[4] Nanda, Neel, et al. "Progress measures for grokking via mechanistic interpretability." arXiv preprint arXiv:2301.05217 (2023).

[5] Barak, Boaz, et al. "Hidden progress in deep learning: Sgd learns parities near the computational limit." Advances in Neural Information Processing Systems 35 (2022): 21750-21764.

**Questions:**

See weaknesses

**Limitations:**

The authors adequately address the limitations.

---

> ### Author Rebuttal · Authors · 2023-08-09
>
> We thank Reviewer ccWP for their time and feedback! Replying to each comment, suggestion and/or question in turn:
>
> > As the authors point out in their related work, the idea that emergence appears due to the metric is not novel, and was proposed by [1], so the novelty of this paper lies in the quantitative evaluation of this hypothesis
>
> Srivastava et al. 2022 indeed questioned whether emergent abilities of LLMs may be due to the metric. However, respectfully, we feel the reviewer undervalues our paper’s novelty & contributions. Our work converts a few sentences into precise conceptual understanding, which in turn generates empirically testable predictions in LLMs, suggests new meta-analyses of previous publications’ results, and inspires new experiments that we implement. Specifically: Firstly, we converted Srivastava et al. 2022’s question into a novel simple mathematical model that is both intuitive and quantitatively testable. Secondly, we used our mathematical model to generate empirically testable predictions and then tested those predictions, confirming them. Thirdly, we proposed and executed a novel meta-analysis of BIG-Bench to reveal additional supporting evidence. Fourthly, we identified a class of metrics that Srivastava et al. did not discuss (i.e., discontinuous metrics such as Multiple Choice Grade). Lastly, we showed how metric choice can induce seemingly emergent abilities in a different modality that had never before been shown, providing an explanation for why emergent abilities have (to the best of our knowledge) not been observed in vision. We also mention additional causes for skepticism, such as previous papers’ possible failure to control for multiple comparisons.
>
> > There has been work on examining emergence and progress measures in similar contexts, e.g. grokking, which should be addressed in the related work [2–5]
>
> We see several significant differences between grokking and emergent abilities of LLMs:
> 1. Grokking is primarily studied within a single model, whereas emergent abilities are studied within a model family
> 2. Grokking occurs with increasing gradient steps, whereas emergent abilities occur with increasing model scale typically measured in parameters or effective parameters (although more recently compute).
> 3. Grokking explicitly studies a discrepancy between the model’s train and test behavior, whereas emergent abilities (to the best of our knowledge) do not present separate train & test curves.
> 4. Grokking is primarily studied on toy tasks in small networks, whereas emergent abilities are studied on benchmark tasks in large language models.
>
> While we personally appreciate the grokking literature, and while we may be mistaken, we respectfully feel that grokking is too distant from this work. If requested, we would be happy to add a comparison with grokking to our Appendix.
>
> > There was no discussion of the relationship to alternative hypotheses. The authors mention in the related work multiple papers that suggest that abilities do, in fact, emerge with scale, but the authors should explain how their hypothesis relates.
>
> Respectfully, we do discuss alternative hypotheses and how they relate to our paper on Lines 209-212: “Caballero et al. [4] explain emergence by assuming a piece-wise power law functional form; under this view, emergent abilities are real, caused by a change in the governing power law. In contrast, our work suggests that emergent abilities are induced by the researcher, even under a single power law. Michaud et al. [25] posit that emergent abilities may be real under strong data assumptions.”
>
> Are you requesting that we discuss the differences in greater depth? We would be happy to expound here, as well as in the paper.
>
> Caballero et al. 2023 contrasts with our work in that their hypothesized explanation for emergent abilities is that the governing power law changes at specific break points. Our explanation for emergent abilities is that the researcher’s analyses create the seemingly emergent abilities. Both explanations could be true: some emergent abilities might genuinely be abruptly appearing, whereas some emergent abilities might be attributable to the metric.
>
> Michaud et al. [25] posits that language modeling data might be comprised of discrete subtasks (“quanta”) that networks learn. Larger networks have greater capacity, and thus are more capable of learning more of these quanta. If some downstream task requires a network to learn some combination of quanta, then larger networks are more likely to have all the requisite capabilities and thus are capable of performing this downstream task. We think that this is a very interesting hypothesis. Whether language modeling data can or should be understood from this quantization perspective, and whether these quanta indeed are the origin of emergent abilities, are really exciting questions that we think merit more study!
>
> > There was no discussion of the BLEU score metric, even though there are many tasks that have a very high score with this metric
>
> Previous papers have not made many claims about emergent abilities under BLEU score; to the best of our knowledge, we know of only one task (IPA transliterate) on which at least one model family exhibits emergent abilities. In contrast, the hand-annotated emergent abilities by https://www.jasonwei.net/blog/emergence almost all seem to occur under other metrics (Multiple Choice Grade, Accuracy, Exact Match). Due to a dearth of claimed emergent abilities under BLEU, we chose to focus on the more prevalent metrics.
>
> That said, we actually had an appendix subsection on BLEU score that we cut due to the relative rareness of BLEU and due to the subsection’s length (explaining BLEU requires more work than Exact Match). We would be happy to put the appendix subsection back in if requested. The answer is that BLEU is also a sharp nonlinear metric.
>
> > Minor: a lot of the figures are small and therefore hard to read
>
> We will fix this. Thank you!

---

> > ### Comment · Reviewer_ccWP · 2023-08-11
> >
> > Thanks for the response.
> >
> > > However, respectfully, we feel the reviewer undervalues our paper’s novelty & contributions.
> >
> > I did not mean to suggest that this paper does not have many valuable contributions!
> >
> > I wanted only to point out that the idea that the fact that emergence-looking behavior can arise from a choice of metric is not new. In particular, in addition to the hypothesis of Srivastava et al. 2022, the work by Nanda et al. 2023 suggests that grokking arises similarly from the choice of metric, and that under a different metric (e.g., a progress measure, as they call it), the discontinuous jump in performance disappears. For this reason, I do believe that the work in grokking is relevant to your work (at least this paper, possibly others).
> >
> > > Are you requesting that we discuss the differences in greater depth? We would be happy to expound here, as well as in the paper.
> >
> > Yes, exactly, thank you for the further discussion.
> >
> > > We would be happy to put the appendix subsection back in if requested. The answer is that BLEU is also a sharp nonlinear metric.
> >
> > My main concern was that in Figure 5, it appears there are many tasks that leverage the BLEU metric with a high emergence score, suggesting that there may be emergence with the BLEU metric. Since there was no discussion of this in the paper, I was wondering how these tasks fit into your argument.
> >
> >
> > Overall, I was mostly concerned with the discussion of the BLEU metric. With some added discussion of the BLEU metric, I think that the authors would have addressed my comments and I would raise my score.

---

> > > ### Author Response · Authors · 2023-08-13
> > > **Response to Reviewer ccWP's Response**
> > >
> > > > I did not mean to suggest that this paper does not have many valuable contributions!
> > >
> > > Ah! We are so sorry - we misunderstood your tone. We apologize for the miscommunication. Thank you for clarifying!
> > >
> > > > For this reason, I do believe that the work in grokking is relevant to your work (at least this paper, possibly others).
> > >
> > > Thank you for clarifying! We now better understand what you meant by progress measures, and what relationship you see. We agree that our perspective is similar to Nanda et al. 2023 and we will add an Appendix section relating emergent abilities with grokking, including appropriate citations to the grokking papers you referenced. We will also mention the Appendix section in our Discussion, but we do not foresee having space to discuss the relationship with grokking at length in our Discussion.
> > >
> > > > Yes, exactly, thank you for the further discussion.
> > >
> > > Sounds good. We will discuss Caballero et al. 2023 and Michaud et al. 2023 at greater length in our Discussion.
> > >
> > > > My main concern was that in Figure 5, it appears there are many tasks that leverage the BLEU metric with a high emergence score, suggesting that there may be emergence with the BLEU metric. Since there was no discussion of this in the paper, I was wondering how these tasks fit into your argument.
> > >
> > > That's a reasonable thought to ponder! To the best of our knowledge, despite our discovery that many BIG-Bench, tasks appear to score highly on the emergence metric under the BLEU metric, we've only seen one claimed BLEU emergent ability on a single task (IPA transliterate). We will explicitly state this in our paper and add our appendix. We will also clarify that the emergence score was introduced by Srivastava et al. 2022 as an approximate proxy for emergent abilities.
> > >
> > > > Overall, I was mostly concerned with the discussion of the BLEU metric. With some added discussion of the BLEU metric, I think that the authors would have addressed my comments and I would raise my score.
> > >
> > > We will also add back in our Appendix section explaining the BLEU metric.
> > >
> > >
> > > **Please consider updating your score if you think our agreed changes merit it!** (FYI: NeurIPS this year doesn't permit uploading a revised manuscript during the review process)

---

> > > > ### Comment · Reviewer_ccWP · 2023-08-15
> > > >
> > > > Any chance you could summarize here the content of the Appendix section you are re-adding? Also, regardless of which tasks have been previously claimed as emergent, it appears that are many tasks that have a high emergence score with the BLEU metric and I was wondering if you could comment on that, as well.

---

> > > > > ### Author Response · Authors · 2023-08-16
> > > > > **Response to Reviewer ccWP**
> > > > >
> > > > > > Any chance you could summarize here the content of the Appendix section you are re-adding?
> > > > >
> > > > > Of course! A number of changes will be made. These include but are not limited to:
> > > > >
> > > > > 1. In the main text, a new Limitations section
> > > > > 2. In the Discussion section, a longer commentary on the relation of our work to alternative hypotheses
> > > > > 3. In the Appendix, a new section "Relationship between Emergent Abilities & Grokking"
> > > > > 4. In the Appendix, a new subsection on BLEU as a metric
> > > > >
> > > > > > Also, regardless of which tasks have been previously claimed as emergent
> > > > >
> > > > > Let us clearly state that we are uncomfortable disregarding previous claims of emergence. Our paper is an investigation of previous claims, and we strongly do not want to invent & misattribute scientific claims to other researchers.
> > > > >
> > > > > Regarding BLEU, to the best of our knowledge, 3 out of 4 papers on emergent abilities didn't mention BLEU, and the 1 paper that does mention BLEU showed BLEU on 1 task (out of many) and did not emphasize the ability. We are hesitant to comment on BLEU because we don't think many authors are claiming emergent abilities under it.
> > > > >
> > > > >
> > > > > > Also, regardless of which tasks have been previously claimed as emergent, it appears that are many tasks that have a high emergence score with the BLEU metric and I was wondering if you could comment on that, as well.
> > > > >
> > > > > We indeed find high emergent scores under BLEU. However, despite this, the authors who studied BIG-Bench (Srivastava et al. 2022) do not claim that these are emergent abilities. We do not know why. Srivastava & colleagues briefly discuss this in their main text, but defer discussion to "Figure App.4" where they switch from BLEU in the main text to ROUGE-L-Sum in Figure App.4's caption. This is why we included our Appendix A4 on ROUGE-L-Sum to demonstrate how it scales. **We will add another appendix A4 discussing how BLEU scales.**
> > > > >
> > > > > Figure 5's X axis ("Emergence Score", called Breakthrough Score in Srivastava et al. 2022) is a rough proxy for emergent abilities. The authors who introduced the score clearly labeled it a heuristic metric. In our view, the Emergence Score should be viewed as "This _might_ merit further investigation" not "These _are_ emergent abilities." In our opinion, this Emergence Score is better thought of as a necessary but insufficient criterion.
> > > > >
> > > > > If you have specific experiments to suggest regarding BLEU, please do let us know!
> > > > >
> > > > > Lest the point be missed, **the key takeaway from Figure 5's left subfigure is that 34 out of 39 BIG-Bench metrics seem to display no possible signs of emergent abilities.**

---

> > > > > > ### Comment · Reviewer_ccWP · 2023-08-18
> > > > > >
> > > > > > Thanks for the response!
> > > > > >
> > > > > > > We are hesitant to comment on BLEU because we don't think many authors are claiming emergent abilities under it.
> > > > > >
> > > > > > I understand that from [33] only IPA transliterate was featured as emergent, but I don't think it's fair to say that authors aren't considering emergent abilities under it, considering that it is featured in their main figure.
> > > > > >
> > > > > > > We indeed find high emergent scores under BLEU. However, despite this, the authors who studied BIG-Bench (Srivastava et al. 2022) do not claim that these are emergent abilities. We do not know why.
> > > > > >
> > > > > > I think the fact that you find especially notable emergent scores under BLEU (and because it is only one of five metrics that you observe high scores on) warrants further investigation into the metric, even if prior work does not properly consider the metric. If you're looking for an experiment, one possible idea would be to reproduce something similar to Figure 9 (in A.4) but for the BLEU metric. If BLEU is a sharp metric, great. Even if it is the case that BLEU is not a sharp metric, I don't think this undermines your argument at all, as I understand your argument to be that the appearance of most *but not necessarily all* emergence arises from a particular choice of metric. Nonetheless, I do think that it's important to discuss BLEU in the paper to answer the immediate question that a reader might have when looking at Figure 5 and observing that the most notable metric (tasks with highest emergence scores) is BLEU, and not the other metrics that you do discuss. Considering how notable BLEU appears in Figure 5, I think it also might be worth including at least some discussion of this in the main paper, in addition to your proposed extension of the appendix.
> > > > > >
> > > > > > Overall, I think conditional on the added discussion of BLEU (regardless of whether BLEU is a sharp metric or not), I have been convinced that this is a solid paper, and so I will increase my score to a 7.

---

### Official Review · Reviewer_eK8N · 2023-07-06

**Soundness:** 3 good
**Presentation:** 4 excellent
**Contribution:** 4 excellent
**Rating:** 8
**Confidence:** 3

**Summary:**

This paper aims to debunk a possibly myth about the emergent properties of language models with regard to scale. The emergent behaviors are defined as abilities that are not demonstrated with smaller models, but emerge suddenly with an increase in the model capacity, and cannot be explained by smoothly extrapolating from the model capacity -- i.e., emergent abilities are sharp and unpredictable. The claim that the authors seek to validate is that emergent abilities are actually artifacts of the fact that the choice of evaluation metric, and that continuous metrics (instead of discrete) show smooth increases in performance with scale -- i.e., that "emergent" abilities are in fact smooth and predictable.

**Strengths:**

This paper has many strengths:
- First off, it is well-written.
- The paper is timely -- large language models have only become moreso in the center of attention in the past year with abundant claims surrounding emergent abilities.
- The claim that discrete metrics are a source of so-called "emergence" of new abilities is reasonable and makes intuitive sense.
- The empirical evaluation is thorough, with evaluations of GPT-3/InstructGPT models, a meta-analysis using BigBench, and using discrete metrics to artificially induce "emergent behavior" in existing vision models.

**Weaknesses:**

I may have misunderstood parts of the argument, but I am not entirely convinced that emergent abilities might not be real, when it is clear that they are real with respect to discrete metrics (that often have practical interpretations). The paper seems to question the basic premise of emergent abilities, i.e., whether or not they are real. The authors cite the following definition early on -- emergent abilities are "abilities that are not present in smaller-scale models but are present in large-scale models; thus they cannot be predicted by simply extrapolating the performance improvements on smaller-scale models,” which itself depends on an evaluation metric. Since the definition depends on an evaluation metric, why does changing the metric make emergent abilities a mirage? Shouldn't emergent abilities only hold, then, with respect to a metric?

Per my understanding of this work, it seems to me that incorrect conclusions have been drawn in prior work by *only* considering discrete evaluation metrics, which is not the full picture -- pointing this out is the main contribution of this paper, and this is an important point to make. However, it still seems that these properties are real with respect to discrete metrics, and these discrete metrics are often desirable. Does this make the choice of metric bad, and emergence a mirage as a result? Or does this render prior analyses incomplete, since they do not consider metrics in which performance **does** smoothly change?

**Questions:**

- Why are smooth evaluation metrics necessarily desirable if they are sometimes a relaxation of the real objective, which may actually be discrete (e.g. the 0-1 loss)?
- NOTE: please switch to the NeurIPS 2023 template, I noticed that you are using the 2022 version of the template.

**Limitations:**

The authors address this.

---

> ### Author Rebuttal · Authors · 2023-08-09
>
> We thank Reviewer eK8N for their time and feedback! Replying to each comment, suggestion and/or question in turn:
>
>
> > The authors cite the following definition early on -- emergent abilities are "abilities that are not present in smaller-scale models but are present in large-scale models; thus they cannot be predicted by simply extrapolating the performance improvements on smaller-scale models,” which itself depends on an evaluation metric. Since the definition depends on an evaluation metric, why does changing the metric make emergent abilities a mirage? Shouldn't emergent abilities only hold, then, with respect to a metric?
>
>
> This is a good question! Our view is that the answer is no. An ability is a talent, skill, or proficiency in a particular area, whereas a metric is one way of quantitatively measuring that ability. Many different metrics can be used for measuring a particular ability. For example, if we want to quantify how good someone is at playing basketball in the NBA, there are many metrics we might turn to: number of won championships, true shooting percentage, effective field goal percentage, player efficiency rating, etc.
>
>
> To explain another way, emergent abilities are discussed as profound changes of a model family with scale. But if indeed fundamental changes are occurring in the model family, then the changes should appear under many metrics. What we show is that this doesn’t seem to appear in the claimed emergent abilities we analyze.
>
>
> > I may have misunderstood parts of the argument, but I am not entirely convinced that emergent abilities might not be real, when it is clear that they are real with respect to discrete metrics (that often have practical interpretations).
>
>
> Respectfully, “real” and “not real” is terminology we try to avoid. Rather, the claim of “emergent abilities” is that as models get larger, their behavior changes sharply and unpredictably. Our paper argues that previously observed sharp and unpredictable changes might be better attributed to the metric than to fundamental changes in the model family with scale. We use the term “mirage” because, from a distance, models appear to be changing abruptly, but upon closer investigation, the abrupt changes can disappear and become more predictable.
>
>
> > However, it still seems that these properties are real with respect to discrete metrics, and these discrete metrics are often desirable.
>
>
> We agree; indeed, we show this in our Figure 3 Row 1! Our paper is not saying previous papers’ experimental results are wrong or that the metrics they use should not be used. Rather, our point is that previous papers might have been surprised by scaling behavior because of the metrics they used or because of limited data. In addition, we show that even for such discrete metrics, the jump is an expected behavior and provide evidence against labeling this as "emergence," i.e., an unpredictable jump. Furthermore, by changing the metric or increasing the resolution using more data, they might have discovered smoother, more predictable trends with scale.
>
>
> > Or does this render prior analyses incomplete, since they do not consider metrics in which performance does smoothly change?
>
>
> This statement is closer to what we have in mind, but we think our paper goes beyond merely saying previous papers are “incomplete”. We think a conceptual blind spot existed in multiple papers that our paper identified. To explain, “Emergent abilities” doesn’t mean that bigger models perform better than smaller models. Rather, “emergent abilities” means that bigger models _suddenly and unexpectedly_ perform better than smaller models. To our knowledge, previous papers did not try changing metrics or collecting more data to increase resolution. Our work qualitatively predicts the effects (and interaction!) that metric and sample size should have, then confirms those predictions.
>
>
> > Does this make the choice of metric bad, and emergence a mirage as a result?
>
> > Why are smooth evaluation metrics necessarily desirable if they are sometimes a relaxation of the real objective, which may actually be discrete (e.g. the 0-1 loss)?
>
>
> Thank you for giving us a chance to clarify! We are not criticizing any metric, nor recommending one metric over another metric. Our main takeaway is that *if* you want to be able to predict the scaling behavior of a model family, the metric you choose may have an effect on your results, and so you should take into account the effect your chosen metric will have so you aren’t surprised. A corollary is that if you want to make accurate scaling forecasts, continuous & linear metrics might be more advantageous, but if you do want to use discontinuous or nonlinear metrics (which is totally fine!), then you may need lots of data to have sufficient resolution to accurately estimate model performance under your chosen metric. You should of course choose whichever metric is best suited for your setting - just make sure to think through what effects your metric choice will have! We will make these points more clearly in our Discussion. Hopefully that helps!
>
>
> > NOTE: please switch to the NeurIPS 2023 template, I noticed that you are using the 2022 version of the template.
>
> Great catch - thank you for letting us know! We will fix this!

---

> > ### Comment · Reviewer_eK8N · 2023-08-13
> >
> > Thank you for the clarifications and helping me to understand what I missed in my review! With these addressed, I'm happy to raise my score to an 8. Great work!

---

### Official Review · Reviewer_GXCv · 2023-07-06

**Soundness:** 4 excellent
**Presentation:** 4 excellent
**Contribution:** 4 excellent
**Rating:** 9
**Confidence:** 4

**Summary:**

This paper shows that the widely-discussed phenomenon of "emergent abilities" in large language models is largely due to the choice of (discontinuous/nonlinear) metrics and underpowered analyses.


**Strengths:**

- Well-written paper studying an important/commonly discussed phenomenon
- Comprehensive, well-designed, and convincing experiments + analysis


**Weaknesses:**

No major weaknesses - the paper was a pleasure to read!


**Questions:**

- l227: Not sure what is meant by "multiple controls" - is the implication that the community should have more aggregate / multi-metric benchmarks like BIG-Bench?
- comment: I found myself wanting to compare Fig 4 side-by-side with Fig 3 to understand the change from additional test data, but it was difficult to do so (especially with the diff y-axis scaling).

**Limitations:**

Yes

---

> ### Author Rebuttal · Authors · 2023-08-09
>
> We thank Reviewer GXCv for their time and feedback, and we’re overjoyed to hear you write that the paper was a pleasure to read! Replying to each comment, suggestion and/or question in turn:
>
> > l227: Not sure what is meant by "multiple controls" - is the implication that the community should have more aggregate / multi-metric benchmarks like BIG-Bench?
>
> To confirm, the sentence you’re referring to is “In this particular setting, emergent abilities claims are possibly infected by a failure to control for *multiple comparisons*”? Assuming the answer is yes, thank you for giving us a chance to clarify! Here, “multiple comparisons” doesn’t refer to aggregate (sometimes called holistic) evaluation like BIG-Bench or HELM.
>
>
> Rather, “multiple comparisons” refers to a phenomenon where someone can draw an incorrect conclusion after considering many attempts because they forget to normalize by the number of attempts. The point is that if one runs many tests, the probability of at least one false positive increases. XKCD #882 explains this through a parable of jelly beans causing acne.
>
> Here, in the context of emergent abilities, the risk is that if one looks at the scaling behavior of 10^6 task-model family-metric triplets, one probably should expect to find at least a few seemingly unpredictable changes due to random chance alone. The solution to multiple comparisons (oftentimes referred to as “controlling for multiple comparisons”) is to raise one’s threshold for declaring a discovery. For instance, we might say, “Because we’re looking at 10^6 task-model family-metric triplets, in order for us to be confident that emergent abilities are a phenomenon (and not just due to random chance), we need to see emergent abilities on at least 1% of the tests.” The statistical hypothesis testing community has ways of making this precise and statistically justified; a good starting point might be the [family-wise error rate](https://en.wikipedia.org/wiki/Family-wise_error_rate) page. Hopefully, this helps clarify!
>
> > comment: I found myself wanting to compare Fig 4 side-by-side with Fig 3 to understand the change from additional test data, but it was difficult to do so (especially with the diff y-axis scaling).
>
> Very understandable! We need the y-axes to have different scaling to make our point that by changing the y-axis and by using more data to increase the resolution, the seemingly zero-accuracy values are actually non-zero accuracies. Would you recommend visualizing the effect of sample size in a different way? For instance, we could show estimation error at different sample sizes by subsampling the larger dataset at different subsample sizes. We have these plots and could submit them as part of the 1 page PDF addendum.

---

> > ### Comment · Reviewer_GXCv · 2023-08-10
> > **Acknowledgment**
> >
> >  I've read the rebuttal response — thanks for answering my questions!

---

### Official Review · Reviewer_Cfbn · 2023-07-24

**Soundness:** 2 fair
**Presentation:** 3 good
**Contribution:** 4 excellent
**Rating:** 7
**Confidence:** 4

**Summary:**

The paper shows that the apparent suddenness with which capabilities of language models seem to emerge with scale may be an artifact of the metrics by which they are measured. Theoretically, emergence should be more gradual if the metric scales linearly with log loss on some test set. Three sources of empirical evidence are used to support this claim. First, the authors study the scaling of the GPT-3 family on arithmetic tasks and show directly that emergence is smoother when a linear metric is used. Second, they conduct a meta-analysis of BIG-Bench, and show that gradual emergence only occurs with non-linear metrics, and that the effect is reduced when switching to a linear metric. Third, they artificially induce sudden emergence by using non-linear metrics for autoencoders on CIFAR100 and for transformers on Omniglot.

**Strengths:**

The basic point made by the paper is an important one that has not been fully appreciated by researchers. It is something that everyone studying scaling laws should understand and incorporate into their future work.

**Weaknesses:**

I think there are notable weaknesses in the empirical evidence given to support the basic point of the paper, and insufficient acknowledgement of the limitations of the evidence and other possibilities that could still give rise to sudden emergence.

On the empirical evidence:
- The main direct evidence that previous claims of sudden emergence are an artifact of the metric comes from studying those previous claims directly, which is done in Section 3. However, the evidence here is somewhat inconclusive: for example for 2-digit multiplication with target str len 4 (which is presumably most 2-digit multiplications?) the jumps from 350M->1.3B->6.7B are gradual while there is a much bigger jump from 6.7B->175B. Moreover, this bigger jump is not predicted by the theoretical model. So it seems like there is still some relatively sudden emergence (even though it is reduced by using the right metric).
- The BIG-Bench analysis is helpful, but again on logical_args it seems like there may still be an unpredictable jump. And on the other tasks, the emergence is not that sudden even with the original metric.
- The induced emergence experiments are also helpful, but they show that sudden emergence can be induced by using a non-linear metric, not that it is always induced by a using a non-linear metric. So this evidence is more indirect.
- Overall I think the paper would have come to more correct (and probably more nuanced) conclusions if it had studied all of the tasks from Figure 1, where sudden emergence had previously been claimed, in detail, instead of studying just one of these tasks combined with other, weaker sources of evidence.
- Nevertheless, for an initial exploration of the topic, I think combining multiple source of evidence, even if somewhat weak, is helpful, and it is reasonable to defer a more careful analysis to future work.

On other theoretical possibilities:
- There is no guarantee that because validation loss scales smoothly, cross-entropy loss on a specific sub-distribution must also scale smoothly. This depends on how the model allocates capacity to that sub-distribution. Therefore there is still an important theoretical possibility of sudden emergence, even with a linear metric. Indeed, this may explain why the data did not match the theoretical model in Section 3.
- In some cases, it may not be possible to construct a linear metric. For example, for program synthesis, whether the program passes the unit tests cannot be easily converted to something like edit distance or log loss.
- It is argued that log accuracy can be used in this case. But this is not necessarily feasible. For example, if current models pass unit tests with probability 1 in a trillion, it is not feasible to measure this accurately enough to perform any useful extrapolation. Therefore there may be sudden emergence for all practical purposes.

Overall, I thought that the evidence was strong enough to support a conclusion that sudden emergence is partially explained by the choice of metric, and that researchers should prefer metrics that scale linearly with log loss when available. I also think that this conclusion is itself enough for the paper to be an important contribution. But I think the paper should take more care not to overstate its claims and acknowledge the limitations of the evidence and, space permitting, other theoretical possibilities. In particular, claims like "metric choice is likely wholly responsible for emergent abilities" (p. 9 line 204) are not justified and should be removed or reworded.

I am recommending acceptance of this paper because of the importance of the contribution, but I think that some effort should be made to revise the paper prior to acceptance. However, I am confident in the ability of the authors to do this.

Some presentation points:
- I think it would be clearer to talk about "sudden emergence" and "gradual emergence" rather than "emergence" and "no emergence", since "emergence" can also be interpreted as "appearing despite not being explicitly trained for". However, I understand that the current terminology is taken from the existing literature and the paper cannot be blamed for this.
- I suggest changing the title of the paper, because the current title can be misinterpreted as asking whether the _abilities_ are mirages, rather than whether the emergence (or sudden emergence as I prefer to put it) is a mirage. Any researcher who reads the abstract should not be misled, but unfortunately provocative titles that can easily be misinterpreted can be picked up on by the media and end up misleading the public. However, I acknowledge that choosing titles is hard and no title is perfect.

More minor points:
- p. 4 line 78 - I am not familiar with this Delta notation (or I have a vague recollection of it). It is clear from the context so probably fine, but consider alternative notation/clarifying the definition.
- p. 5 line 115 - Why choose this task from among all the tasks in Figure 1, and why 2-shot / 4-digit? (Would be even nicer to include results on other tasks/other numbers of digits, but absent that, it would be good to clarify how you chose, to make it clear it was not cherry-picked.)
- Figure 3: Target str len 1: how do you get a 1-digit answer from multiplying two 2-digit numbers? I assume you are counting things like "01 * 01"? Maybe clarify.
- Figure 3: I think it would be clearer to include titles at the top of each column saying "Mathematical Model", "2-Integer 2-Digit Multiplication Task", "2-Integer 4-Digit Addition Task", instead of putting this in the caption. That way people can easily see at a glance what each plot is showing, without having to dig into the caption.
- Figure 4: Same comment.
- Figure 5: It would be nicer to group the metrics based on their mathematical form. E.g. it is unclear what "gender_bias_score" is. This also might help spot if there are more examples with emergence that didn't have high emergence score.
- Figure 5B is a bit redundant. Maybe just point out in figure 5A which of the 5 did not count as emergence. It's also a bit unclear why this didn't count despite the emergence score.
- p. 9 line 209 - I don't think this work really contradicts broken neural scaling laws. Even if sudden emergence goes away when you change the metric, one might still be interested in forecasting the original metric.

**Questions:**

No questions other than those given above

**Limitations:**

No limitations other than those given above

---

> ### Author Rebuttal · Authors · 2023-08-09
>
> Thanks for time & feedback! You made clear effort to think critically about how to improve our paper! 6k space limited -> dense response; try our best.
>
> > Why few-shot int arithmetic
>
> 1. Few-shot int arithmetic featured prominently in multiple prev papers L102-104.
> 2. To make point about sufficient resolution, needed to be able to generate more samples; generating samples easier for arithmetic than other tasks, e.g., Persian QA. We’ll add.
>
> > Better to study all tasks from Fig 1
>
> Agreed but not fully feasible: 4/5 models private (LaMDA, Gopher, Chinchilla, PaLM). PaLM2 recently became queryable, but to best of knowledge, PaLM 1 spanning several scales is not. Before submitting, tried obtaining networks’ outputs via priv comms with authors but unsuccessful. Thanks to your comment, we realized we can study GPT-3 fam on IPA Transliterate & Grounded Mappings. Aren’t sure about MMLU since MMLU=57 tasks; may be beyond our budget. How important would collect+analyze IPA Transliterate & Grounded Mappings be?
>
>
> > main direct evidence [...] is somewhat inconclusive
>
> Our work follows ethos: all models are wrong, but some are useful. Role of mathematical model: explain quantitatively what scaling behavior we should qualitatively expect in LLMs & why. As stated, math model relies on approximations (per-token error rates are approximately constant and independent) that are only ~true. Despite, math model usefully predicts effects of: varying length of target string, changing metric, & sampling additional data to incr resolution.
>
> > [BIG-Bench] logical_args seems like there may still be unpredictable jump
>
> > And on the other tasks, emergence is not that sudden even with original metric.
>
> BIG-Bench’s data sometimes confusing. Example: Swahili English Proverbs (red): Multiple Choice Grade improved for largest model, but Brier Score regressed; inconsistent? Data from private model (LaMDA) -> unable to investigate. Unsatisfying answer is claims of emergent abilities involve subjective judgment.
>
> > The induced emergence experiments are also helpful, but [...] this evidence is more indirect.
>
> Agreed. Point: show that metric choices can make well-known networks appear to possess emergent abilities on commonplace tasks.
>
> > There is no guarantee that because validation loss scales smoothly, cross-entropy loss on a specific sub-distribution must also scale smoothly [...] Therefore there is still an important theoretical possibility of sudden emergence
>
> Agree. L218: “We emphasize that nothing in this paper should be interpreted as claiming that large language models _cannot_ display emergent abilities.” **Will add Limitations section** including this. Interesting hypotheses exist & are cited e.g., Michaud et al. 2023.
>
> > It may not be possible to construct a linear metric, e.g., program synthesis
>
> Indeed, linear metric not always possible, but to clarify, we aren’t advocating linear metrics, nor criticizing any metrics. Main point: **if** goal to predict scaling behavior of model family, chosen metric may have effect, so take that effect into account so you aren’t surprised. Corollary: to make accurate scaling forecasts, continuous & linear metrics are probably more advantageous, but if discontinuous & nonlinear metrics preferred, then may need lots of data to have sufficient resolution to accurately measure performance. Choose whichever metric is best for your setting - just make sure to think through consequences of metric choice!
>
> Linear metrics might be more broadly applicable; Program Synthesis: count number of unit tests passed e.g. HackerRank coding interviews
>
> > [log accuracy] is not necessarily feasible.
>
> Not claiming log accuracy necessarily feasible. Rather, claim: if one wishes to use a particular metric when forecasting scaling behavior, then one should consider whether they possess sufficient data to have sufficient resolution to accurately estimate performance under their metric. In some cases, this might be feasible, but in other cases, this might not be. Will clarify.
>
> > Therefore there may be sudden emergence for all practical purposes.
>
> This involves a hidden step that we think is unlikely. We are skeptical that accuracy could transition from ~1e-12 to ~1 without detectably passing through intermediate orders of magnitude at intermediate model scales. While not impossible, to best of knowledge, such an abrupt transition has not been observed.
>
> > the paper should take more care not to overstate its claims and acknowledge the limitations
>
> Agreed. Will add Limitations sec & replace strong claims e.g., “metric choice is likely wholly responsible for emergent abilities” with measured statements e.g., “metric choice might possibly be responsible for some claimed emergent abilities.”
>
> > p. 4 line 78
>
> Delta = prob simplex. Will clarify.
>
> > p. 9 line 209 - doesn’t contradict broken neural scaling laws
>
> Don’t contradict, but dissimilar. BNSL attributes emergent abilities to changepoint in scaling behavior -> $\geq$2 scaling law vs  our paper says _even with 1 scaling law_, emergent abilities might be due to the metric.
>
> > suggest changing the title of the paper
>
> > clearer to talk about "sudden" & "gradual emergence"
>
> Terminology tricky & attempts resulted in dissatisfied colleagues. Difficulty with “sudden” vs “gradual emergence” is emergence has rich history in physics, typically at phase transitions; “gradual emergence” could be viewed as oxymoron. We have also been told not to use “phase transition;” one colleague: “Unless you can write down a partition function and show that it's non-analytic, I strongly suggest not doing this.” Also tried “Emergent Abilities” in quotation marks to make clear referencing prior terminology; quotation marks interpreted as mocking prior work. Have more stories. After trying repeatedly, we feel current terminology is best available compromise. Tempering our claims & adding Limitations sec will help. Happy to hear additional suggestions.
>
> Other sugg changes will be made.

---

> > ### Comment · Reviewer_Cfbn · 2023-08-12
> > **Response to rebuttal**
> >
> > Thank you for the rebuttal. I am glad to hear that we are in agreement on several points. I continue to think that this is an important contribution, but also very hard to get comprehensive data on. While additional data would always help, I think the existing data is enough for the current paper and it's more important to be upfront about the limitations of the evidence as it stands. Therefore I am pleased to hear that a limitations section will be added.
> >
> > On the question of accuracy transitioning from ~1e-12 to ~1, I agree with the authors here. I was being a bit hyperbolic, but I think my point still stands: it's not so unusual for a 10x increase in model size (i.e. 100x increase in training compute, by [1]) to improve accuracy by ~30x (see for example 2-Integer 2-Digit Multiplication in the paper). Since compute spending is growing rapidly (see for example [2], which suggests a 10-month doubling time), that means that an ability could go from essentially undetectable (say 1e-3 accuracy, at which level it's hard to tell if it's a fluke) to useful performance (say 1e-1) over the course of a few years, or perhaps faster in some cases. (This isn't really a technical disagreement, it's more a question of how the result that log accuracy scales smoothly should be framed: I'm saying that even if this is true, we should still expect to be surprised by future model capabilities.)
> >
> > [1] https://arxiv.org/abs/2203.15556
> >
> > [2] https://epochai.org/blog/compute-trends

---

> > > ### Author Response · Authors · 2023-08-13
> > > **Response to Reviewer Cfbn's Comment**
> > >
> > > Phew! We're so sorry for the dense earlier message and hope that it all made sense! A 6k character response is such a tight limit when a reviewer like yourself provides so much good feedback.
> > >
> > > >  I am glad to hear that we are in agreement on several points.
> > >
> > > Indeed! We agreed with you on almost all points. I think that being more upfront about our paper's limitations by adding an explicit Limitations section will be an important addition.
> > >
> > > > I continue to think that this is an important contribution, but also very hard to get comprehensive data on.
> > >
> > > As a personal comment, an inability to obtain more comprehensive data has been one of the most disheartening aspects of working on this topic. We want to have a better scientific understanding of emergent abilities, but typically, models aren't public, nor are models' text outputs, and obtaining that data ourselves is slow and expensive.
> > >
> > > >  I'm saying that even if this is true, we should still expect to be surprised by future model capabilities.
> > >
> > > Agreed.
> > >
> > > Once again, thank you so much for your comments & feedback earlier! It was quite obvious to us that you thought deeply about how our paper could be improved, and we very much value your perspective. We will update our manuscript accordingly.
> > >
> > > **Please consider updating your score if you think our agreed changes merit it!**  (FYI: NeurIPS this year doesn't permit uploading a revised manuscript during the review process)

---

### Decision · Program_Chairs · 2023-09-21

**Decision:**

Accept (oral)

**Comment:**

This work provides a closer look at the nature of increased capabilities of machine learning models at scale, arguing that prior evidence for unpredictable and qualitative leaps in capability is confounded by the metrics employed to measure such capabilities. This has broad implications on how we describe and evaluate the capabilities of model.

Reviewers found the support provided for this claim to be thorough and appreciated the additional nuance resulting from the discussion period. The critical perspective provided by this work is timely and will be of broad interest at the conference, and also just provides an important lesson.